SciPost Physics

# Time-reversal symmetry breaking and resurrection in driven-dissipative Ising models

Daniel A. Paz[1*], Mohamamd F. Maghrebi[1]

**1** Department of Physics and Astronomy, Michigan State University, 426 Auditorium Rd. East Lansing, MI 48823, USA
* pazdanie@msu.edu

June 23, 2021

## Abstract

Fluctuation-dissipation relations (FDRs) and time-reversal symmetry (TRS), two pillars of statistical mechanics, are both broken in generic driven-dissipative systems. These systems rather lead to non-equilibrium steady states far from thermal equilibrium. Driven-dissipative Ising-type models, however, are widely believed to exhibit *effective* thermal critical behavior near their phase transitions. Contrary to this picture, we show that both the FDR and TRS are broken even macroscopically at, or near, criticality. This is shown by inspecting different observables, both even and odd operators under time-reversal transformation, that overlap with the order parameter. Remarkably, however, a modified form of the FDR as well as TRS still holds, but with drastic consequences for the correlation and response functions as well as the Onsager reciprocity relations. Finally, we find that, at criticality, TRS remains broken even in the weakly-dissipative limit.

# 1  Introduction

Quantum systems in or near equilibrium define a paradigm of modern physics. The past two decades, however, have witnessed a surge of interest in non-equilibrium systems thanks to the advent of novel experimental techniques where quantum matter is observed far from equilibrium. An immediate challenge is that general guiding principles of equilibrium statistical mechanics are not directly applicable in this new domain. One such general feature is the principle of detailed balance in equilibrium [1]. Extensions of this principle to the quantum domain have been studied extensively for both closed and open systems [2–5]. In all such settings, detailed balance is directly tied to time-reversal symmetry (TRS) under reversing the direction of time (in two-time correlators, e.g.). A second defining characteristic of equilibrium systems is the fluctuation-dissipation relations (FDRs) relating the dynamical response of the system to their inherent fluctuations. Importantly, these two principles are not independent: a proper formulation of the TRS leads to the FDRs [6].

A generic non-equilibrium setting is defined by driven-dissipative systems characterized by the competition of an external drive and dissipation due to coupling to the environment. This competition leads the system towards a non-equilibrium steady state far from thermal equilibrium [7, 8]. Due to the non-equilibrium dissipative dynamics, both TRS and FDR are generally broken in these steady states [9]; the guiding principles of equilibrium physics are thus absent in their driven-dissipative counterparts. Nonetheless, it has become increasingly clear that the critical properties of a large class of many-body driven-dissipative systems (yet not all [10–14]) are described by an effective equilibrium behavior near their respective phase transitions [15–27]. This is particularly the case for Ising-like phase transitions where the order parameter takes a relatively simple form and the dynamics is rather constrained. Thermal critical behavior is even observed in the driven-dissipative Dicke phase transition [28].

In this work, we consider driven-dissipative Ising-type models, but, contrary to what is generally believed, we show that both FDR and TRS are broken even macroscopically at or near criticality. This is shown by inspecting different observables that overlap with the order parameter and crucially encompass both even and odd operators under time-reversal transformation. We show that these observables satisfy emergent FDR-like relations but with effective temperatures that are opposite in sign; we dub such relations FDR*. Moreover, while TRS is broken macroscopically, we show that a modified form of the time-reversal symmetry of two-time correlators, dubbed TRS*, emerges at or near criticality where correlation and response functions exhibit definite, but possibly opposite, parities under time-reversal transformation. This is in sharp contrast with equilibrium where correlation and response functions exhibit the same parity.

We showcase our results in the context of two relatively simple models, enabling exact analytical and numerical calculations. The main model considered here is an infinite-range driven-dissipative Ising model, a descendant of the paradigmatic open Dicke model [29,30]. We also consider a short-range quadratic model of driven-dissipative bosons with the Ising symmetry. These models provide an ideal testbed for the general questions about the fate of the FDR and TRS in driven-dissipative systems, the role of the time-reversal symmetry (breaking), and the emergence of modified fluctuation-dissipation relations.

We begin by summarizing our main results in Section 2. Building on the techniques developed in a recent work [31], we set up a non-equilibrium field theory and calculate the exact correlation and response functions in Section 3. In Section 4, we determine the effective temperatures and provide evidence for the modified FDR* and TRS* via exact analytics and numerics. We further show that, even in the limit of vanishing dissipation, the TRS breaking or restoring depends on a certain order of limits. In Section 5, we present an effective field theory from which we prove the FDR* and TRS* and furthermore derive the modified Onsager reciprocity relations. Finally, in Section 6, we make the case for the broader application of our conclusions in the setting of a short-range driven-dissipative model of coupled bosons.

## 2 Main Results

Characteristic information about a given system and a set of observables $\hat{O}_i$ can be obtained from the two-point functions

$$C_{O_i O_j}(t) = \langle \{\hat{O}_i(t), \hat{O}_j\} \rangle, \quad \chi_{O_i O_j}(t) = -i\Theta(t)\langle [\hat{O}_i(t), \hat{O}_j] \rangle, \tag{1}$$

which define the correlation function and the causal response function, respectively; the former captures fluctuations (e.g., at equal times), while the latter describes the response of the system to a perturbation at an earlier time. Fluctuation-dissipation theorem, a pillar of statistical mechanics, relates these two at equilibrium. For our purposes, we write the fluctuation-dissipation relation (FDR) as [32]

$$\text{FDR}: \qquad \chi_{O_i O_j}(t) = \frac{1}{2T}\Theta(t)\partial_t C_{O_i O_j}, \tag{2}$$

valid for classical systems as well as quantum systems at finite temperature and at long times [32]. Furthermore, if the system satisfies *microreversiblity*, or (quantum) detailed balance, two-time correlators exhibit a time-reversal symmetry [3,4]. Assuming that the operator $\hat{O}_i$ has a definite parity $\epsilon_i$ under time-reversal (in the absence of magnetic fields), the correlation and response functions then satisfy [33]

$$C_{O_i O_j}(t) = \epsilon_i \epsilon_j C_{O_j O_i}(t), \tag{3a}$$

$$\chi_{O_i O_j}(t) = \epsilon_i \epsilon_j \chi_{O_j O_i}(t). \tag{3b}$$

In this work, we shall refer to such relations as TRS of two-time correlators, or just TRS. Notice that these set of equations are also consistent with the FDR in Eq. (2). The above equations form the origin of the Onsager reciprocity relations [34].

FDR and TRS are both broken in driven-dissipative systems as they give rise to a non-equilibrium steady states at long times. Extensive effort has gone into identifying the steady states of many-body driven-dissipative systems as well as their phase transitions. A large body of work, however, has shown that a variety of driven-dissipative many-body systems exhibit critical behavior that is *effectively* equilibrium [15–28]. Specifically, an effective temperature $T_{\text{eff}}$ emerges that governs the critical properties (e.g., critical exponents) near their phase transitions at long times/wavelengths. An effective TRS may be then expected to emerge as well given that TRS and FDR are intimately tied [6].

In this work, we consider driven-dissipative systems whose Hamiltonian—in the rotating frame—is itself time-reversal symmetric: $\hat{T}\hat{H}\hat{T}^{-1} = \hat{H}$ with $\hat{T}$ the antiunitary operator associated with the time-reversal transformation; here, $\hat{T} = K$ is simply complex conjugation. Dissipative coupling to the environment, however, explicitly breaks TRS and exposes

the non-equilibrium nature of the system. Additionally, we assume that the full dynamics under the Liouvillian $\mathcal{L}$ comes with an Ising $\mathbb{Z}_2$ symmetry that defines the order parameter at the phase transition. Non-equilibrium systems with the $\mathbb{Z}_2$ symmetry are generally expected to fall under the familiar Ising universality class at their phase transitions. In fact, it is known that the Ising universality class is robust against non-equilibrium perturbations [35]. In harmony with this picture, previous work on driven-dissipative Ising-type systems has reported an emergent FDR governing the order-parameter dynamics for some $T_{\text{eff}}$ [17, 20, 28, 36–38].

Notwithstanding the evidence for emergent equilibrium, here we report that FDR and TRS are both macroscopically broken in driven-dissipative Ising-type systems. This becomes manifest by considering other observables that overlap with the order parameter, i.e., observables that share the same $\mathbb{Z}_2$ symmetry. In the Ising model, for example, beside $\hat{S}_x$ typically signifying the order parameter, we will also consider $\hat{S}_y$ (with the transverse field along the $z$ direction). This expanded set of observables exhibit critical scaling, but they do not obey an effective FDR. Interestingly, however, we show that a modified form of the FDR emerges as

$$\text{FDR}^* : \qquad \chi_{O_i O_j^*}(t) \simeq \frac{1}{2 T_{\text{eff}}} \Theta(t) \partial_t C_{O_i O_j}, \tag{4}$$

up to noncritical corrections; we dub this modified relation FDR*. Here, we have assumed that the $\hat{O}_i$'s are Hermitian operators[1], and defined $\hat{O}_j^* = \hat{T} \hat{O}_j \hat{T}^{-1}$ (recall that $\hat{T} = K$). In the example of the Ising model, $\hat{S}_x^* = \hat{S}_x$ while $\hat{S}_y^* = -\hat{S}_y$. The FDR* is radically different from its equilibrium counterpart, and has important consequences. To see this, let us again assume that the operator $\hat{O}_i$ has a definite parity $\epsilon_i$ under time-reversal transformation. In this case, the FDR can be written as

$$\chi_{O_i O_j}(t) \simeq \frac{\epsilon_j}{2 T_{\text{eff}}} \Theta(t) \partial_t C_{O_i O_j}. \tag{5}$$

This means that an emergent FDR is satisfied with $\chi_{O_i O_j} = (1/2T_{ij}) \partial_t C_{O_i O_j}$ but with different temperatures for different observables, $T_{ij} = \epsilon_j T_{\text{eff}}$, same in magnitude but possibly with opposite signs depending on the observables. For example, if $\hat{O}_1$ is even under time-reversal ($\epsilon_1 = 1$) and $\hat{O}_2$ is odd ($\epsilon_2 = -1$), we find $T_{11} = -T_{12} = T_{21} = -T_{22} = T_{\text{eff}}$.

We further show that an unusual form of TRS holds at or near criticality:

$$C_{O_i O_j}(t) \simeq C_{O_j O_i}(t), \tag{6a}$$

$$\chi_{O_i O_j}(t) \simeq \epsilon_i \epsilon_j \chi_{O_j O_i}(t). \tag{6b}$$

In parallel with FDR*, the above relations will be referred to as TRS*. Notice that the above equations are consistent with the FDR* in Eq. (5). Interestingly, the correlation and response functions transform differently under time-reversal transformation, in sharp contrast with equilibrium; cf. Eq. (3). While violating TRS, these functions still have a definite parity under time-reversal transformation. Moreover, combining Eqs. (5) and (6), we further show that the Onsager reciprocity relation finds a modified form with the opposite parity. This is surprising in light of the broken TRS, but is a direct consequence of the emergent TRS*.

We derive these results via a simple field-theoretical analysis that identifies a slow mode in the vicinity of the phase transition. We show that the FDR* and TRS* are a consequence of the non-Hermitian form of the dynamics generator, due to the TRS of the Hamiltonian, $\hat{T} \hat{H} \hat{T}^{-1} = \hat{H}$, combined with the Ising $\mathbb{Z}_2$ symmetry of the Liouvillian $\mathcal{L}$.

---

[1] Unlike the standard FDR, the FDR* is sensitive to the operators being Hermitian or not; see Section 6.2.

# 3 Driven-Dissipative Ising Model

Here, we briefly introduce the infinite-ranged driven-dissipative Ising model with spontaneous emission (DDIM) [31]. The DDIM describes a system of $N$ driven, fully-connected 2-level atoms under a transverse field, and subject to individual atomic spontaneous emission. In the rotating frame of the drive, the Hamiltonian is given by

$$\hat{H} = -\frac{J}{N}\hat{S}_x^2 + \Delta\hat{S}_z \,, \tag{7}$$

with $J$ an effective Ising coupling and $\Delta$ the transverse field. For clarity, we use the total spin operators $\hat{S}_\alpha = \sum_i \hat{\sigma}_i^\alpha$, with $\hat{\sigma}^\alpha$ the usual Pauli matrices. The Markovian dynamics of the system is given by the quantum master equation [39]

$$\frac{d\hat{\rho}}{dt} = \mathcal{L}[\hat{\rho}] = -i[\hat{H}, \hat{\rho}] + \Gamma \sum_i \hat{\sigma}_i^- \hat{\rho}\hat{\sigma}_i^+ - \frac{1}{2}\{\hat{\sigma}_i^+\hat{\sigma}_i^-, \hat{\rho}\} \,. \tag{8}$$

Here, $\hat{\rho}$ is the reduced density matrix of the system and the (curly) brackets represents the (anti)commutator. The first term on the RHS generates the usual quantum coherent dynamics, while the remaining terms describe the spontaneous emission of individual atoms at a rate $\Gamma$. While there is no time dependence in the rotating frame, detailed balance is directly broken, and the model is indeed non-equilibrium [40]. Equation (8) exhibits a $\mathbb{Z}_2$ symmetry ($\hat{\sigma}_i^{x,y} \to -\hat{\sigma}_i^{x,y}$), which is spontaneously broken in the phase transition from the normal phase ($\langle\hat{S}_{x,y}\rangle = 0$) to the ordered phase ($\langle\hat{S}_{x,y}\rangle \neq 0$). Due to the collective interaction, the DDIM phase diagram is exactly obtained via mean field theory in the thermodynamic limit. However, it is also a physically relevant model, realized experimentally either in the large-detuning limit of the celebrated open Dicke model [31, 41–46], or directly through trapped-ions [47]. At the same time, this model allows for exact analytical and numerical calculations, and provides an ideal testbed for our conclusions. We will also consider a short-range model in Section 6 where we arrive at the same conclusions.

In the models considered in this paper, the operator $\hat{T} = K$ is simply complex conjugation. The Hamiltonian in Eq. (7) is time-reversal symmetric because it is real. Time-reversal transformation (i.e., acting with the anti-unitary operator $\hat{T}$ together with sending $t \to -t$) leaves the von Neumann equation $\partial_t\hat{\rho} = -i[\hat{H}, \hat{\rho}]$ invariant. In the ground state, this symmetry enforces $\langle\hat{S}_y\rangle = 0$ as $\hat{T}\hat{S}_y\hat{T}^{-1} = -\hat{S}_y$; this is true even in the ordered phase where $\langle\hat{S}_y\rangle = 0$ while $\langle\hat{S}_x\rangle \neq 0$ [38]. Furthermore, correlators such as $\langle\{\hat{S}_x, \hat{S}_y\}\rangle$ that are odd under time-reversal must be zero. More generally, these symmetry considerations can be extended to thermal states under unitary dynamics as they satisfy the KMS condition and exhibit an equilibrium symmetry that involves time-reversal [48, 49]. Two time-correlators then satisfy the symmetry relations in Eq. (6). However, the driven-dissipative model in Eq. (8) breaks such symmetries. This is because Eq. (8) is derived in the rotating frame of the drive, hence breaking detailed balance. The resulting steady state is then *not* a thermal state, and TRS of two-time correlators no longer holds [50]. Specifically, this allows for nonzero expectation values of odd observables such as $\langle\hat{S}_y\rangle$ (in the ordered phase) and correlators such as $\langle\{\hat{S}_x, \hat{S}_y\}\rangle$.

Despite the infinite-range nature of the model, individual atomic dissipation makes the problem nontrivial since the total spin is no longer conserved. To make analytical progress, we adopt the approach that we have developed in Ref. [31]. We provide the technical steps in the following subsections; a non-technical reader may wish to skip ahead to Section 4 for the relevant results.

## 3.1 Non-equilibrium field theory

Using a non-equilibrium quantum-to-classical mapping introduced in Ref. [31, 38], we can map exactly the non-equilibrium partition function (normalized to unity)

$$Z = \text{Tr} \left( \hat{\rho}_{\text{ss}} \right) = \lim_{t \to \infty} \text{Tr} \left( e^{t\mathcal{L}} \hat{\rho}(0) \right) = 1, \tag{9}$$

to a Keldysh path-integral over a pair of real scalar fields, representing the order parameter of the phase transition. We have introduced the steady state density matrix $\rho_{\text{ss}}$, defined as the long-time limit of the Liouvillian dynamics governed by Eq. (8). The process is done by vectorizing the density matrix, such that the non-equilibrium partition function takes the form

$$Z = \langle\langle I | e^{t\mathbb{L}} | \rho_{\text{ss}} \rangle\rangle, \tag{10}$$

where we have performed the transformation $|i\rangle\langle j| \to |i\rangle \otimes |j\rangle$ on the basis elements of the density matrix. The matrix $\mathbb{L}$ is given by

$$\mathbb{L} = -i \left( \hat{H} \otimes \hat{I} - \hat{I} \otimes \hat{H} \right) + \Gamma \sum_i \left[ \hat{\sigma}_i^- \otimes \hat{\sigma}_i^- - \frac{1}{2} \left( \hat{\sigma}_i^+ \hat{\sigma}_i^- \otimes \hat{I} + \hat{I} \otimes \hat{\sigma}_i^+ \hat{\sigma}_i^- \right) \right]. \tag{11}$$

Following the vectorization procedure, we perform a quantum-to-classical mapping via a Suzuki-Trotter decomposition in the basis that diagonlizes the Ising interaction, and then utilize the Hubbard-Stratonovich transformation on the (now classical) collective Ising term [31, 38]. Tracing out the leftover spin degrees of freedom leaves us with a path-integral representation of the partition function:

$$Z = \int \mathcal{D}[m_c(t), m_q(t)] e^{i\mathcal{S}[m_{c/q}(t)]}, \tag{12}$$

with the Keldysh action

$$\mathcal{S} = -2JN \int_t m_c(t) m_q(t) - iN \ln \text{Tr} \left[ \mathcal{T} e^{\int_t \mathbb{T}(m_{c/q}(t))} \right], \tag{13}$$

where $\mathcal{T}$ is the time-ordering operator. For convenience, we have introduced the *classical* and *quantum* Hubbard-Stratonovich fields $m_{c/q}$ in the usual Keldysh basis [19, 51]. The order parameter $\langle \hat{S}_x \rangle$ is given by the average of $m_c$ which takes a non-zero expectation value in the ordered phase; correlators involving the operator $\hat{S}_x$ too can be directly written in terms of the fields $m_{c/q}$ [31]. The matrix $\mathbb{T}$ in Eq. (13) in the basis defined by $\hat{\sigma}^x \otimes \hat{I}$ and $\hat{I} \otimes \hat{\sigma}^x$, where

$$\hat{\sigma}^x = \begin{pmatrix} 1 & 0 \\ 0 & -1 \end{pmatrix}, \quad \hat{\sigma}^y = \begin{pmatrix} 0 & i \\ -i & 0 \end{pmatrix}, \quad \hat{\sigma}^z = \begin{pmatrix} 0 & 1 \\ 1 & 0 \end{pmatrix}, \tag{14}$$

is given by

$$\mathbb{T} = \begin{pmatrix} -\frac{\Gamma}{4} + i2\sqrt{2}Jm_q & i\Delta & -i\Delta & \frac{\Gamma}{4} \\ i\Delta - \frac{\Gamma}{2} & -\frac{3\Gamma}{4} + i2\sqrt{2}Jm_c & -\frac{\Gamma}{4} & -i\Delta - \frac{\Gamma}{2} \\ -i\Delta - \frac{\Gamma}{2} & -\frac{\Gamma}{4} & -\frac{3\Gamma}{4} - i2\sqrt{2}Jm_c & i\Delta - \frac{\Gamma}{2} \\ \frac{\Gamma}{4} & -i\Delta & i\Delta & -\frac{\Gamma}{4} - i2\sqrt{2}Jm_q \end{pmatrix}. \tag{15}$$

Note the overall factor of $N$ in Eq. (13) due to the collective nature of the Ising interaction, meaning that the saddle-point approximation is exact in the limit that $N \to \infty$. We mention here that Eq. (13) indeed describes the steady state of the quantum master equation in Eq. (8), arising due to the competition between drive and dissipation.

## 3.2 Correlation and response functions

This action is only in terms of the scalar fields $m_{c/q}$, which are related to the observable $\hat{S}_x$ [31]. To obtain the correlation and response functions for $\hat{S}_y$ and the cross-correlations with $\hat{S}_x$, we introduce source fields $\alpha^{(u/l)}$ and $\beta^{(u/l)}$ to Eq. (11) which couple to $\hat{S}_x$ and $\hat{S}_y$ respectively:

$$\mathbb{L}'(t) = \mathbb{L} + i\alpha^{(u)}(t)\frac{\hat{S}_x}{\sqrt{N}} \otimes \hat{I} - i\alpha^{(l)}(t)\hat{I} \otimes \frac{\hat{S}_x}{\sqrt{N}} + i\beta^{(u)}(t)\frac{\hat{S}_y}{\sqrt{N}} \otimes \hat{I} + i\beta^{(l)}(t)\hat{I} \otimes \frac{\hat{S}_y}{\sqrt{N}}, \quad (16)$$

and perform the non-equilibrium quantum-to-classical mapping as usual. The absence of a minus sign on the last term stems from the vectorization transformation in the mapping. Introducing the sources does not affect the quadratic term in $m$ in Eq. (13), but changes the $\mathbb{T}$ matrix to the new matrix $\mathbb{T}' = \mathbb{T} + \mathbb{T}_\alpha + \mathbb{T}_\beta$ where

$$\mathbb{T}_\alpha = i\sqrt{\frac{2}{N}} \begin{pmatrix} \alpha_q & 0 & 0 & 0 \\ 0 & \alpha_c & 0 & 0 \\ 0 & 0 & -\alpha_c & 0 \\ 0 & 0 & 0 & -\alpha_q \end{pmatrix}, \quad (17)$$

and

$$\mathbb{T}_\beta = \frac{1}{\sqrt{2N}} \begin{pmatrix} 0 & -\beta_c + \beta_q & -\beta_c - \beta_q & 0 \\ \beta_c - \beta_q & 0 & 0 & -\beta_c - \beta_q \\ -\beta_c - \beta_q & 0 & 0 & -\beta_c + \beta_q \\ 0 & -\beta_c - \beta_q & \beta_c - \beta_q & 0 \end{pmatrix}. \quad (18)$$

We have performed the Keldysh rotation $\alpha_{c/q} = (\alpha^{(u)} \pm \alpha^{(l)})/\sqrt{2}$, $\beta_{c/q} = (\beta^{(u)} \pm \beta^{(l)})/\sqrt{2}$ for convenience. Next, we expand the action to quadratic order in both $x_{c/q}$ and the source fields around $m_{c/q} = \alpha_{c/q} = \beta_{c/q} = 0$,

$$\mathcal{S} = \frac{1}{2}\int_{t,t'} \begin{pmatrix} m_c \\ m_q \\ \alpha_c \\ \alpha_q \\ \beta_c \\ \beta_q \end{pmatrix}_t^T \begin{pmatrix} \mathbf{P} & 0 & 0 \\ 4J\mathbf{P}_{\alpha\alpha} & \mathbf{P}_{\alpha\alpha} & 0 \\ 4J\mathbf{P}_{\beta\alpha} & 2\mathbf{P}_{\beta\alpha} & \mathbf{P}_{\beta\beta} \end{pmatrix}_{t-t'} \begin{pmatrix} m_c \\ m_q \\ \alpha_c \\ \alpha_q \\ \beta_c \\ \beta_q \end{pmatrix}_{t'}, \quad (19)$$

where the kernel becomes a lower triangular block matrix. The block matrices take the usual Keldysh structure

$$\mathbf{P} = \begin{pmatrix} 0 & P^A \\ P^R & P^K \end{pmatrix}, \qquad \mathbf{P}_{\alpha\alpha} = \frac{1}{4J^2}\left[\mathbf{P} + \begin{pmatrix} 0 & 2J\delta(t) \\ 2J\delta(t) & 0 \end{pmatrix}\right],$$

$$\mathbf{P}_{\beta\alpha} = \begin{pmatrix} 0 & P_{\beta\alpha}^A \\ P_{\beta\alpha}^R & P_{\beta\alpha}^K \end{pmatrix}, \qquad \mathbf{P}_{\beta\beta} = \mathbf{P}_{\alpha\alpha},$$

and the matrix elements for each block matrix are

$$P^R(t) = P^A(-t) = -2J\delta(t) + \Theta(t)8J^2 e^{-\frac{\Gamma}{2}t}\sin(2\Delta t), \quad (20a)$$

$$P^K(t) = i8J^2 e^{-\frac{\Gamma}{2}|t|}\cos(2\Delta t), \quad (20b)$$

$$P_{\beta\alpha}^R(t) = -P_{\beta\alpha}^A(-t) = -\Theta(t)2e^{-\frac{\Gamma}{2}|t|}\cos(2\Delta t), \quad (20c)$$

$$P_{\beta\alpha}^K(t) = -i2e^{-\frac{\Gamma}{2}|t|}\sin(2\Delta t)\,, \tag{20d}$$

Equation (19) is exact in the thermodynamic limit, as higher-order terms in the expansion are at least of the order $\mathcal{O}(1/N)$.

After Fourier transformation, defined as $m(t) = \int_\omega e^{-i\omega t} m(\omega)$ with the integration measure $\int_\omega = \int_{-\infty}^\infty d\omega/2\pi$, we integrate out the $m_{c/q}$ fields to obtain the generating functional $W[\alpha_{c/q}, \beta_{c/q}] = -i\ln Z$ as

$$W = -\frac{1}{2}\int_\omega \begin{pmatrix}\alpha_q\\\beta_q\\\alpha_c\\\beta_c\end{pmatrix}_\omega^\dagger \begin{pmatrix}\mathbf{G}^K & \mathbf{G}^R\\ \mathbf{G}^A & 0\end{pmatrix}_\omega \begin{pmatrix}\alpha_q\\\beta_q\\\alpha_c\\\beta_c\end{pmatrix}_\omega. \tag{21}$$

The Green's function block matrices are given by

$$\mathbf{G}^K = \begin{pmatrix} G_{xx}^K & G_{xy}^K\\ G_{yx}^K & G_{yy}^K\end{pmatrix}, \qquad \mathbf{G}^R = \begin{pmatrix} G_{xx}^R & G_{xy}^R\\ G_{yx}^R & G_{yy}^R\end{pmatrix}, \tag{22}$$

and satisfy $\mathbf{G}^K(\omega) = -[\mathbf{G}^K]^\dagger(\omega)$ and $\mathbf{G}^R(\omega) = [\mathbf{G}^A]^\dagger(\omega)$. In terms of the original observables $\hat{S}_x, \hat{S}_y$, the Green's functions become $G_{jj'}^K(\omega) = -i\mathcal{F}_\omega\langle\{\hat{S}_j(t), \hat{S}_{j'}(0)\}\rangle/N$ and $G_{jj'}^R(\omega) = -i\mathcal{F}_\omega\Theta(t)\langle[\hat{S}_j(t), \hat{S}_{j'}(0)]\rangle/N$. The elements of Eq. (22) are given by

$$G_{xx}^K(\omega) = \frac{-i\Gamma[\Gamma^2 + 4(4\Delta^2 + \omega^2)]}{2(\omega - \omega_1)(\omega - \omega_2)(\omega - \omega_1^*)(\omega - \omega_2^*)}\,, \tag{23a}$$

$$G_{xy}^K(\omega) = \frac{4\Gamma(iJ\Gamma + 2J\omega - 2\Delta\omega)}{(\omega - \omega_1)(\omega - \omega_2)(\omega - \omega_1^*)(\omega - \omega_2^*)}\,, \tag{23b}$$

$$G_{yy}^K(\omega) = \frac{-i\Gamma[\Gamma^2 + 16(2J - \Delta)^2 + 4\omega^2]}{2(\omega - \omega_1)(\omega - \omega_2)(\omega - \omega_1^*)(\omega - \omega_2^*)}\,, \tag{23c}$$

$$G_{xx}^R(\omega) = \frac{4\Delta}{(\omega - \omega_1)(\omega - \omega_2)}\,, \tag{23d}$$

$$G_{xy}^R(\omega) = \frac{\Gamma - 2i\omega}{(\omega - \omega_1)(\omega - \omega_2)}\,, \tag{23e}$$

$$G_{yx}^R(\omega) = \frac{-\Gamma + 2i\omega}{(\omega - \omega_1)(\omega - \omega_2)}\,, \tag{23f}$$

$$G_{yy}^R(\omega) = \frac{-4(2J - \Delta)}{(\omega - \omega_1)(\omega - \omega_2)}\,, \tag{23g}$$

where $\omega_{1/2} = -\frac{i}{2}(\Gamma \mp \Gamma_c)$, $\Gamma_c = 4\sqrt{\Delta(2J - \Delta)}$.

## 4   Non-Equilibrium Signatures

In this section, we discuss the macroscopic, critical behavior of the driven-dissipative Ising model introduced in Eq. (8). It is generally believed that such Ising models find an emergent equilibrium behavior near their phase transition. This is often argued by considering a single observable such as the order parameter and showing that it satisfies an effective FDR [17, 20, 36–38]. In contrast, we consider different observables and show that

the associated FDR and TRS are both violated even macroscopically. However, we show that a modified form of these relations emerge, dubbed as the FDR* and TRS*, which govern the critical behavior of this system. In the following subsections, we derive the effective temperatures for different set of observables, discuss the breaking and emergence of (modified) TRS, and finally discuss our results in the limit of vanishing dissipation.

## 4.1 Effective temperature

At thermal equilibrium and at low frequencies, the FDR in frequency space can be written as

$$G_{ij}^R(\omega) - G_{ij}^A(\omega) = \frac{\omega}{2T} G_{ij}^K(\omega) \,. \tag{24}$$

To compare against the FDR in the time domain, we identify $C_{O_i O_j}(t) \equiv i G_{ij}^K(t)$ and $\chi_{O_i O_j}(t) \equiv G_{ij}^R(t)$.[2] The above equation follows from another version of the FDR given by [32]

$$-i\chi_{O_i O_j}''(t) = \frac{1}{4T} \partial_t C_{O_i O_j}(t) \,, \tag{25}$$

where $\chi_{O_i O_j}''(t) \equiv \frac{1}{2}\langle[\hat{O}_i(t), \hat{O}_j]\rangle = \frac{1}{2i}\big(G_{ij}^R(t) - G_{ij}^A(t)\big)$; the retarded and advanced Green functions are defined directly from the operators as $G_{ij}^{R/A}(t) \equiv \mp i\Theta(\pm t)\langle[\hat{O}_i(t), \hat{O}_j]\rangle$. While Eq. (2) is restricted to $t > 0$, the above equation is valid at all $t$, making it more suitable for the transition to Fourier space, i.e., Eq. (24). Of course, the two (causal and non-causal) versions of the FDR are equivalent in equilibrium.

Equation (24) has been extensively used to identify an effective temperature even for non-equilibrium systems [31, 36, 37]. In the non-equilibrium setting of our model, however, we would immediately run into a problem for $i \neq j$ when the corresponding operators have different parities under time-reversal transformation (e.g., $T_{\text{eff}}$ becomes infinite or complex valued). To see why, let us anticipate that the TRS* relations reported in Eq. (5) indeed hold, a fact that we will later justify near criticality and at long times. It is then easy to see that $C_{O_i O_j}(t) = C_{O_j O_i}(-t) \simeq C_{O_i O_j}(-t)$ while $\chi_{O_i O_j}''(t) = -\chi_{O_j O_i}''(-t) \simeq -\epsilon_i \epsilon_j \chi_{O_i O_j}(-t)$. Now for two distinct operators $\hat{O}_i$ and $\hat{O}_j$ where $\epsilon_i = -\epsilon_j$, we find that both $C_{O_i O_j}(t)$ and $\chi_{O_i O_j}(t)$ are even in time (for a fixed set of operators). However, this is not compatible with Eq. (25) as it requires $C_{O_i O_j}(t)$ and $\chi_{O_i O_j}(t)$ to have opposite parities. Postulating an effective FDR in this case, valid for all $t$, forces us to include a sign function, $\text{sgn}(t)$, that is, we should substitute $\chi_{O_i O_j}''(t) = \big(G_{ij}^R(t) - G_{ij}^A(t)\big)/2i \to \text{sgn}(t)\chi_{O_i O_j}''(t) = \big(G_{ij}^R(t) + G_{ij}^A(t)\big)/2i$ on the left hand side of Eq. (25) when $\epsilon_i = -\epsilon_j$. Notice that the extended FDR is consistent with the causal FDR in Eq. (2) when $t > 0$, but is now conveniently valid at all times. This extension is informed by the anticipated form of the TRS* which we will justify later. The fluctuation-dissipation relation is now conveniently cast in frequency space: for arbitrary operators $\hat{O}_i$ and $\hat{O}_j$ (with $i$ and $j$ being the same or distinct), the updated FDR takes the form

$$G_{ij}^R(\omega) - \epsilon_i \epsilon_j G_{ij}^A(\omega) = \frac{\omega}{2T_{ij}(\omega)} G_{ij}^K(\omega) \,, \tag{26}$$

where we have now allowed for a frequency- and operator-dependent effective temperature $T_{ij}(\omega)$. It is now clear that, while for $\epsilon_i = \epsilon_j$ the above equation recovers the structure of the FDR (cf. Eq. (24)), a different combination, $G_{ij}^R(\omega) + G_{ij}^A(\omega)$, appears on the left hand side when $\epsilon_i = -\epsilon_j$. The above equation can be brought into a more compact version again by anticipating the TRS* in Eq. (6b) to write $\epsilon_i \epsilon_j G_{ij}^A(\omega) \simeq G_{ji}^R(\omega)$. Utilizing the

---

[2]We are including a normalization factor $1/N$ in the definition of correlation and response functions for convenience.

relation $G_{ji}^A(\omega) = G_{ij}^R(\omega)^*$, we are finally in a position to write an equation for the effective temperature in the low-frequency limit:

$$T_{ij} = \lim_{\omega \to 0} \frac{\omega}{2} \frac{G_{ij}^K}{G_{ij}^R(\omega) - G_{ji}^A(\omega)} = \lim_{\omega \to 0} \frac{\omega}{4} \frac{-iG_{ij}^K}{\operatorname{Im} G_{ij}^R(\omega)} \,. \tag{27}$$

We have taken the low-frequency limit appropriate near criticality. Again we stress that the above equation is consistent with the standard form of the effective FDR for $i = j$, and it correctly incorporates the TRS* for $i \neq j$ with opposite parities.

A shorter, but perhaps less physically motivated, route to the above equation is to start directly from the causal form of the FDR in Eq. (2). The Fourier transform of this equation is given by [33]

$$\chi_{O_i O_j}(\omega) = \frac{1}{2T} \left[ \mathrm{P} \int \frac{d\omega'}{2\pi} \frac{\omega'}{\omega - \omega'} C_{O_i O_j}(\omega') - \frac{i\omega}{2} C_{O_i O_j}(\omega) \right] \,, \tag{28}$$

where P stands for the principal part. Here too, we shall assume the TRS* in Eq. (6a): with $C_{O_i O_j}(t) \simeq C_{O_j O_i}(t) = C_{O_i O_j}(-t)$ regardless of the operators' parities, the correlation function $C_{ij}(t)$ is even in time, hence its Fourier transform, $C_{O_i O_j}(\omega)$, is purely real. Taking the imaginary part of the above equation then yields $\operatorname{Im} \chi_{O_i O_j}(\omega) = -(\omega/4T)C_{O_i O_j}(\omega)$ where $T$ has to be identified with the effective temperature $T_{ij}(\omega)$. Therefore, we arrive at the same definition of the effective temperature in Eq. (27).

Using Eq. (27), we can now identify the effective temperature in the driven-dissipative Ising model (defining $i, j \in \{x, y\}$)

$$T_{xx} = \frac{\Gamma^2 + 16\Delta^2}{32\Delta} \,, \tag{29a}$$

$$T_{yy} = \frac{\Gamma^2 + 16(\Delta - 2J)^2}{32(\Delta - 2J)} \,, \tag{29b}$$

$$T_{xy} = -T_{yx} = \frac{-2J\Gamma^2}{\Gamma^2 + 16\Delta(2J - \Delta)} \,. \tag{29c}$$

These expressions are calculated everywhere in the normal phase and generally take different values (see also [37]), underscoring the non-equilibrium nature of the model at the microscopic level. Equations (29a) and (29b) display non-analytic behaviour, though in different regions of the phase diagram. $T_{xx}$ diverges when $\Delta \to 0$, in agreement with Ref. [52] that reports an infinite temperature in the $\hat{\sigma}^x$ basis. In contrast, $T_{yy}$ diverges when $\Delta = 2J$ for any finite value of $\Gamma$. This divergence coincides with the change in the dynamical behaviour from overdamped to underdamped dynamics as pointed out in [31]. Finally, $T_{xy} = -T_{yx}$ are everywhere finite but opposite for the opposite order of the observables; this is tied to the TRS* as we will discuss later.

The definition of the low-frequency effective temperature is particularly motivated near the phase boundary where there exists a soft mode [31]. Interestingly, at (or near) the phase transition, we find

$$T_{xx} = -T_{xy} = T_{yx} = -T_{yy} = J. \tag{30}$$

Remarkably, these effective temperatures find the same magnitude, but possibly with different signs. While focusing on a single observable (say $\hat{S}_x$) and its dynamics, one might be led to conclude that the system is in effective equilibrium. However, a different observable (say $\hat{S}_y$) exhibits the opposite effective temperature. Notice that all correlation functions (involving $\hat{S}_x$ and/or $\hat{S}_y$) are divergent at the phase transition, i.e., they are all

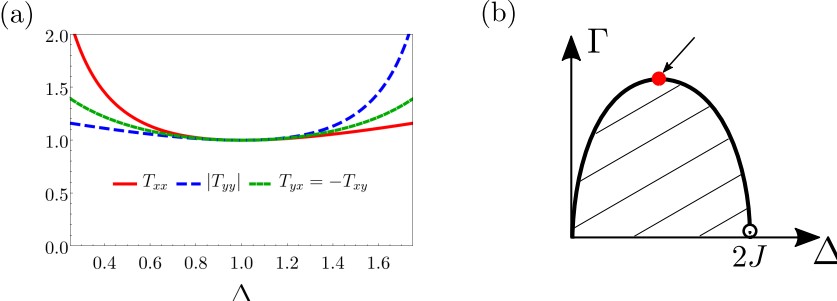

Figure 1: (a) Effective temperatures $T_{xx}$, $T_{xy}$, $T_{yx}$ and $T_{yy}$ as a function of $\Delta$ at or away from the phase boundary; we choose the parameters $J = 1, \Gamma = 4$ with the point $\Delta = 1$ representing the critical point at the tip of the phase boundary (see the red dot in panel (b)). The effective temperatures become equal up to a sign at the critical point. The same pattern emerges on any point along the phase boundary and away from $\Gamma = 0$. (b) The phase diagram of the DDIM. The shaded region is the ordered phase where $\langle \hat{S}_{x,y} \rangle \neq 0$.

sensitive to the soft mode; we will make this more precise in Section 5 where we develop an effective field theory. This suggests that although the critical behavior is governed by a single (soft) mode at the transition, the system is genuinely non-equilibrium even *macroscopically*.

To support these analytical results, we have numerically simulated [38] the FDR in the time domain (cf. Eq. (2)) and at a representative critical point on the phase boundary for a finite, yet large, system with $N = 100$ spins. Correlation and response functions at criticality and at a finite system size require an analysis beyond the quadratic treatment presented here and thus serves as a nontrivial check of our results. Also, working in the time domain and restricting to $t > 0$, we circumvent the issues that arise in the frequency domain; see the discussion in the beginning of this subsection. Indeed, we find an excellent agreement in Fig. 2 between the analytical results (in frequency space) and the numerical results (in the time domain) with the exception of short time differences; the contrast at short times is a consequence of the fact that the (observable-dependent) effective temperature is defined in the zero-frequency limit relevant to the long-time dynamics.

## 4.2 TRS breaking

As discussed in Section 3, broken TRS allows for nonzero correlators such as $\langle \{\hat{S}_x, \hat{S}_y\} \rangle$ that are otherwise odd under the time-reversal transformation. Indeed, we find that this correlator is nonzero and is even critical. More precisely, we find from Eq. (23b) that

$$C_{xy}(t) \equiv iG^K_{xy}(t) = \frac{4}{\Gamma_c} \left[ \frac{-J\Gamma + \text{sgn}(t)(J - \Delta)(\Gamma - \Gamma_c)}{\Gamma - \Gamma_c} e^{-\frac{\Gamma - \Gamma_c}{2}|t|} \right.$$
$$\left. - \frac{-J\Gamma + \text{sgn}(t)(J - \Delta)(\Gamma + \Gamma_c)}{\Gamma + \Gamma_c} e^{-\frac{\Gamma + \Gamma_c}{2}|t|} \right]. \tag{31}$$

(For ease of notation, we have replaced $C_{S_iS_j}$ by $C_{ij}$; similarly for $\chi_{ij}$.) Specifically, at equal times, we have $C_{xy}(t = 0) = -8J\Gamma/(\Gamma^2 - \Gamma_c^2)$. Indeed, the equal-time cross correlation diverges as $\sim 1/(\Gamma - \Gamma_c)$ upon approaching the critical point $\Gamma \to \Gamma_c$. This is a stark manifestation of broken TRS at a macroscopic level. We also note that both $C_{xx}, C_{yy} \sim 1/(\Gamma - \Gamma_c)$ diverge in a similar fashion. Again, this is because $\hat{S}_x$ and $\hat{S}_y$ share the same soft mode, as will be shown in Section 5.

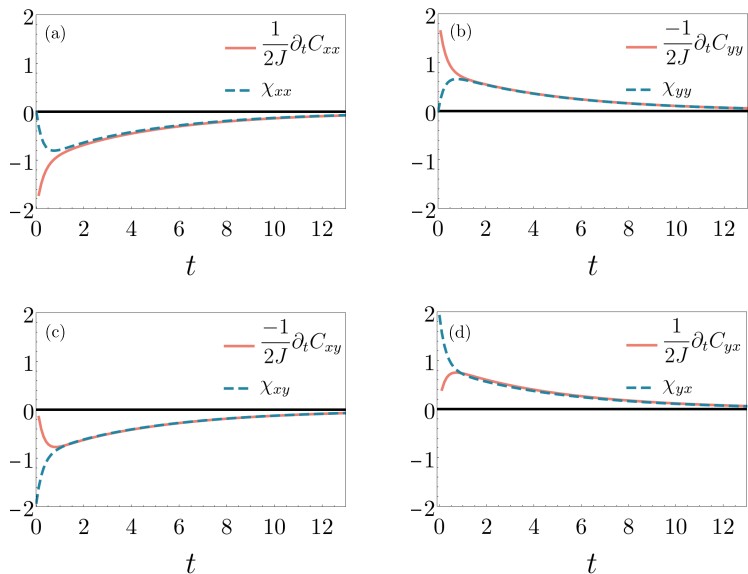

Figure 2: Numerical plots of correlation and response functions at a representative critical point with $J = 1, \Delta = 1, \Gamma = 4$ and the system size $N = 100$. A modified fluctuation-dissipation relation, $\chi_{O_i O_j}(t) = \Theta(t) C_{O_i O_j}(t)/2T_{ij}$, emerges at long times. The effective temperatures take the same value up to a sign: $T_{xx} = -T_{xy} = T_{yx} = -T_{yy} = J$.

The macroscopic breaking of TRS alters the Onsager symmetry relations in an exotic fashion that is distinct for the correlation and response functions. Indeed, the analytical expression in Eq. (31) shows that, near criticality and at sufficiently long times,

$$C_{xy}(t) \simeq -\frac{4J\Gamma}{\Gamma_c(\Gamma - \Gamma_c)} e^{-\frac{\Gamma - \Gamma_c}{2}|t|}, \tag{32}$$

hence, $C_{xy}(t) \simeq C_{xy}(-t)$, or equivalently, $C_{xy}(t) \simeq C_{yx}(t)$ up to noncritical corrections; far from criticality, the correlation functions do not generally satisfy this symmetry relation. Furthermore, the analytical expressions for the response functions in Eqs. (23e) and (23f) show that $\chi_{xy}(t) = -\chi_{yx}(t)$. Interestingly, the cross-correlation and -response functions exhibit opposite parities. These analytical considerations are further supported by the numerical simulation shown in Fig. 3 at criticality confirming

$$C_{xy}(t) \simeq C_{yx}(t), \tag{33a}$$

$$\chi_{xy}(t) \simeq -\chi_{yx}(t), \tag{33b}$$

consistent with the TRS* in Eq. (6). Despite the broken TRS, the correlation and response functions retain definite, though distinct, parities under time-reversal.

## 4.3 Weakly-dissipative limit

In this section, we briefly consider a special limit of the driven-dissipative Ising model, namely a weakly-dissipative critical point at at $\Delta \to 2J$ and $\Gamma \to 0$; see Fig. 1(b). It was shown in previous work that this limit leads to a different critical dynamics than a generic critical point at finite $\Gamma$ [31, 38]. Here, we are interested in the TRS breaking and its possible emergence in the limit of vanishing dissipation. Interestingly, we find that the fate of the TRS depends on the way that this critical point is approached. We shall consider two different scenarios below.

(a)
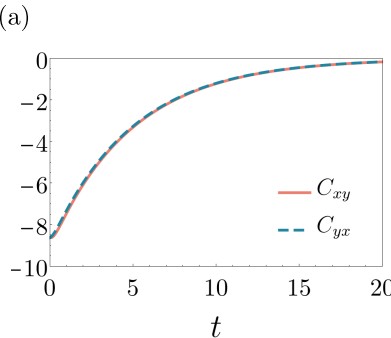

(b)
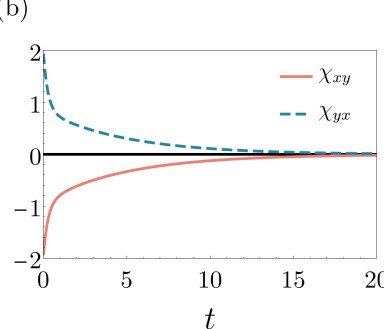

Figure 3: Cross-correlation and -response functions at criticality ($J = 1, \Delta = 1, \Gamma = 4, N = 100$). A modified form of TRS emerges at criticality where correlation (panel a) and response (panel b) functions exhibit opposite parities under time-reversal transformation.

In the first scenario, let us set $\Delta = 2J$ and take the limit $\Gamma \to 0$. Fourier transforming Eq. (23c) to the time domain gives

$$C_{yy}(t) = \lim_{\Delta \to 2J} iG_{yy}^K(t) = 2e^{-\frac{1}{2}\Gamma|t|} . \tag{34}$$

We thus see that $\hat{S}_y$ correlator is finite at the weakly-dissipative critical point, indicating that $\hat{S}_y$ has become "gapped". This appears to suggest a return to the equilibrium scenario where $\hat{S}_y$ plays no role in critical behaviour. However, the cross-correlation given by Eq. (31) remains nonzero and even *critical* at the weakly-dissipative point: $C_{xy}(t = 0) \sim 1/\Gamma$. Therefore, even in the limit of vanishing dissipation, TRS is macroscopically broken.

In the second scenario, we consider $\Delta > 2J$ and first take the limit $\Gamma \to 0$. In this case, we have $\Gamma_c = i\sqrt{\Delta(\Delta - 2J)} \equiv i\omega_c$, which then leads to

$$\lim_{\Gamma \to 0} C_{xy}(t) = \frac{-4(\Delta - J)}{\omega_c} \sin \frac{\omega_c t}{2} . \tag{35}$$

This expression goes to zero at $t = 0$ for any value of $\Delta$ including the weakly dissipative critical point as $\Delta \to 2J^+$, recovering the equilibrium result.

The different behavior in the two scenarios lies in the fact that the system has a finite dissipative gap when we send $\Gamma \to 0$ before sending $\Delta \to 2J$ but not *vice versa*. It has been shown that the steady state of a system with a finite dissipative gap becomes purely a function of the Hamiltonian in the limit of vanishing dissipation, i.e., $\hat{\rho}_{\text{ss}} = f(\hat{H})$ [53]; see also [54]. In this case, the steady state for our model can be written as a function of the Hamiltonian in Eq. (7), and thus respects TRS. This argument however fails in a *gapless* system corresponding to the first scenario considered above. Indeed, we find that in this case the TRS is macroscopically broken even in the limit of vanishing dissipation.

One can also determine the behavior of the effective temperature at the weakly-dissipative critical point. However, since the operator $\hat{S}_y$ is gapped, the definition of the low-frequency effective temperature doesn't seem appropriate. In fact, one finds that the effective temperatures involving this operator take different values (and even diverge) depending on the order of limits. Therefore, we will not report the effective temperature in this limit.

## 5 Effective Field Theory

In this section, we develop a simple, generic field-theory analysis that elucidates the origin of the effective temperatures and their signs as well as FDR* and TRS*. We first need

to construct an action that maps the spin operators $\hat{S}_x$ and $\hat{S}_y$ to the fields $x(t)$ and $y(t)$, respectively. This is achieved by starting from the generating functional $W$ in Eq. (21) and constructing a quadratic action in terms of $x$ and $y$ fields that exactly reproduces the correlations of the corresponding operators. This is simply done via a Hubbard-Stratonovich transformation on $\exp(iW[\alpha_{c/q}, \beta_{c/q}])$ as

$$e^{iW} = \int \mathcal{D}[x_{c/q}, y_{c/q}] e^{i\mathcal{S}_{\text{eff}}[x_{c/q}, y_{c/q}] + i \int_\omega j^T(-\omega)v(\omega)} , \tag{36}$$

where we have absorbed an unimportant normalization factor into the measure, and we have defined the source and field vectors $j = (\alpha_q, \beta_q, \alpha_c, \beta_c)^T$ and $v = (x_c, y_c, x_q, y_q)^T$. The resulting action is given by

$$\mathcal{S}_{\text{eff}} = \frac{1}{2} \int_\omega v^\dagger(\omega) \begin{pmatrix} \mathbf{0} & \mathbf{D}^A \\ \mathbf{D}^R & \mathbf{D}^K \end{pmatrix}_\omega v(\omega) , \tag{37}$$

where we have written the kernel in terms of $2 \times 2$ block matrices:

$$\mathbf{D}^R(\omega) = [\mathbf{D}^A]^T(-\omega) = \begin{pmatrix} 2J - \Delta & \frac{1}{4}(\Gamma - 2i\omega) \\ \frac{1}{4}(-\Gamma + 2i\omega) & -\Delta \end{pmatrix} , \quad \mathbf{D}^K(\omega) = i\frac{\Gamma}{2} \begin{pmatrix} 1 & 0 \\ 0 & 1 \end{pmatrix} . \tag{38}$$

By inspecting the form of $\mathbf{D}^R$, we can identify the soft mode. At the critical point $(\Gamma \to \Gamma_c \equiv 4\sqrt{\Delta(2J - \Delta)})$, this matrix takes the form

$$\mathbf{D}_{\text{cr}}^R(\omega = 0) = \begin{pmatrix} 2J - \Delta & \sqrt{\Delta(2J - \Delta)} \\ -\sqrt{\Delta(2J - \Delta)} & -\Delta \end{pmatrix} . \tag{39}$$

A convenient decomposition of $\mathbf{D}_{\text{cr}}^R(\omega = 0)$ is given by $\mathbf{D}_{\text{cr}}^R(\omega = 0) = \mathbf{U}\mathbf{\Lambda}\mathbf{U}$ where

$$\mathbf{U} = \frac{1}{\sqrt{2J\Delta}} \begin{pmatrix} \Delta & -\frac{1}{4}\Gamma_c \\ \frac{1}{4}\Gamma_c & \Delta \end{pmatrix} , \quad \mathbf{\Lambda} = \begin{pmatrix} 0 & 0 \\ 0 & -2J \end{pmatrix} , \tag{40}$$

valid for $0 < \Delta < 2J$; the regime $\Delta > 2J$ needs to be dealt with separately. The matrix $\mathbf{U}$ is orthogonal, i.e., $\mathbf{U}\mathbf{U}^T = \mathbf{I}$. Notice that this decomposition can be viewed as an SVD where $\mathbf{D}_{\text{cr}}^R(\omega = 0) = \mathbf{U}\mathbf{\Lambda}\mathbf{V}^T$, where $\mathbf{V} = \mathbf{U}^T$ with both $\mathbf{U}$ and $\mathbf{V}$ being orthogonal matrices. In this sense, the left and right vectors are rotated with respect to the original directions in opposite directions; see Fig. 4(a). As we shall see, this is the reason behind the new FDR* and TRS*. This decomposition allows us to express both classical and quantum components of $\phi, \zeta$ in terms of $x, y$ as

$$\begin{pmatrix} \phi_c \\ \zeta_c \end{pmatrix} = \mathbf{U} \begin{pmatrix} x_c \\ y_c \end{pmatrix} = \frac{1}{\sqrt{2J\Delta}} \begin{pmatrix} \Delta x_c - \frac{1}{4}\Gamma_c y_c \\ \frac{1}{4}\Gamma_c x_c + \Delta y_c \end{pmatrix} , \tag{41a}$$

$$\begin{pmatrix} \phi_q \\ \zeta_q \end{pmatrix} = \mathbf{U}^T \begin{pmatrix} x_q \\ y_q \end{pmatrix} = \frac{1}{\sqrt{2J\Delta}} \begin{pmatrix} \Delta x_q + \frac{1}{4}\Gamma_c y_q \\ -\frac{1}{4}\Gamma_c x_q + \Delta y_q \end{pmatrix} . \tag{41b}$$

We note that the diagonal elements of $\mathbf{\Lambda}$ define the masses of the fields $\phi$ and $\zeta$ on the phase boundary. Therefore, we can identify $\phi$ as the soft mode and $\zeta$ as the gapped field. In addition, the Keldysh element of the kernel remains unchanged, $\mathbf{U}^T\mathbf{D}^K\mathbf{U} = \mathbf{D}^K$.

## 5.1 FDR* and TRS*

The field-theory representation makes the origin of the results shown in Section 4 clear. The effective temperatures corresponding to different set of operators can be expressed in terms of $\phi$ and $\zeta$. At the phase boundary, the effective temperature is captured purely by

(a)                                                          (b)

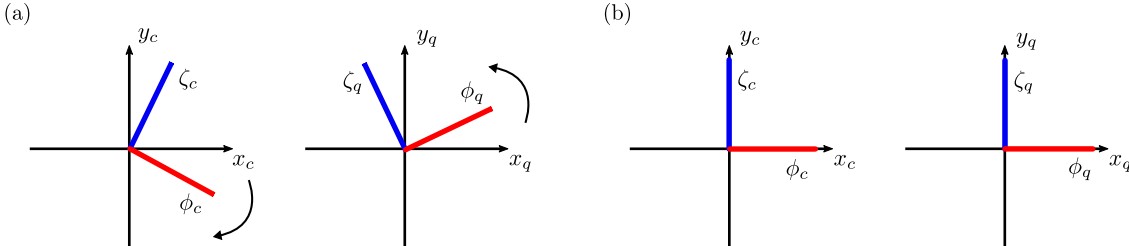

Figure 4: Schematic representation of the massless and massive fields $\phi$ and $\zeta$ in terms of the $x$ and $y$ fields that represent $\hat{S}_x$ and $\hat{S}_y$. (a) The gapped/gapless fields are shown at a generic critical point. The classical and quantum fields are rotated with respect to the $x$-$y$ axes, but in opposite directions, a fact that leads to the opposite signs of the effective temperatures. (b) At the weakly-dissipative critical point, $\Delta = 2J$, $\Gamma \to 0$, the gapless and gapped fields align with the $x$ and $y$ axes, respectively, similar to thermal equilibrium.

the soft mode $\phi$. Therefore, in the low-frequency limit, the dominant contribution to the effective temperature is dictated by the correlation and response functions of $\phi$, while $\zeta$ gives a noncritical correction. We have, up to noncritical corrections,

$$T_{xx} \simeq T_\phi, \tag{42a}$$

$$T_{xy} \simeq \frac{U_{12}}{U_{21}} T_\phi = -T_\phi, \tag{42b}$$

$$T_{yx} \simeq T_\phi, \tag{42c}$$

$$T_{yy} \simeq \frac{U_{12}}{U_{21}} T_\phi = -T_\phi, \tag{42d}$$

where

$$T_\phi \equiv \lim_{\omega \to 0} \frac{\omega}{2} \frac{\langle \phi_c(\omega)\phi_c(-\omega)\rangle}{\langle \phi_c(\omega)\phi_q(-\omega)\rangle - \langle \phi_q(\omega)\phi_c(-\omega)\rangle}, \tag{42e}$$

can be viewed as the effective temperature of the soft mode. This interesting result is purely a consequence of the non-Hermitian structure of Eq. (39). Technically, one can see that the same pattern of effective temperatures emerges whenever the inverse retarded Green's function $\mathbf{D}_0 \equiv \mathbf{D}_{\text{cr}}^R(\omega = 0)$ obeys the relation

$$\tau^z \mathbf{D}_0 \, \tau^z = \mathbf{D}_0^T, \quad \text{with} \quad \tau^z = \begin{pmatrix} 1 & 0 \\ 0 & -1 \end{pmatrix}, \tag{43}$$

which simply states that the off-diagonal part of the matrix $\mathbf{D}_0$ is antisymmetric. Note that $\mathbf{D}_0$ is real, but non-Hermitian. The fact that the kernel $\mathbf{D}_0$ satisfies the above property can be argued solely on the grounds that the Hamiltonian itself is time-reversal symmetric. To show this, let us assume the contrary, namely that the off-diagonal part of the matrix $\mathbf{D}_0$ has a symmetric component. This would give rise to a coupling $\sim x_c y_q + x_q y_c$ where the fields' time dependence is implicit. Breaking the classical and quantum fields in terms of the fields on the forward and backward branches of the Keldysh contour [51], such coupling becomes $\sim x_+ y_+ - x_- y_-$. This term takes the structure of a Hamiltonian contribution to the action $(\mathcal{S}_+ - \mathcal{S}_-)$; however, the Hamiltonian does not couple $x$ and $y$ since it is time-reversal invariant. We should then conclude that the off-diagonal part of $\mathbf{D}_0$ is antisymmetric. In equilibrium, the off-diagonal terms are simply zero (at $\omega = 0$); however, in a driven-dissipative system, dissipation naturally gives rise to nonzero (though antisymmetric) off-diagonal matrix elements.

We remark that a generalized version of the FDR,

$$\mathbf{G}^K(\omega) = \mathbf{G}^R(\omega)\mathbf{F}(\omega) - \mathbf{F}(\omega)\mathbf{G}^A(\omega) \,,$$

is also utilized in the literature [17,19] to determine the distribution function matrix $\mathbf{F}(\omega)$. While in thermal equilibrium $\mathbf{F}(\omega) = \coth(\omega/2T)\mathbf{I}$ is proportional to the identity, the distribution function is allowed to become a nontrivial matrix in driven-dissipative systems, specifically in the context of the open Dicke model (possessing the same symmetries as those considered here). For the the cavity mode, it was shown that this matrix finds two eigenvalues, $\pm\lambda(\omega)$, whose low-frequency behaviour is given by $\lambda(\omega) \sim 2T_{\text{eff}}/\omega$ [17]. The positive eigenvalue was then identified as the effective temperature. In contrast, our analysis clarifies the interpretation of the negative effective temperature in the form of the FDR* in Eq. (4) and its origin due to time-reversal symmetry breaking.

Next, we derive the TRS* relations for the correlation and response functions in Eq. (6), namely $C_{xy}(t) \simeq C_{yx}(t)$ while $\chi_{xy}(t) \simeq -\chi_{yx}(t)$ up to noncritical corrections. Again, the key is to keep only the critical contributions in terms of $\phi_{c/q}$. The symmetry of the correlation function follows in a simple fashion as

$$C_{xy}(t) = 2\langle x_c(t)y_c(0)\rangle \simeq 2U_{11}U_{12}\langle\phi_c(t)\phi_c(0)\rangle \simeq 2\langle y_c(t)x_c(0)\rangle = C_{yx}(t). \qquad (44)$$

For the response function, we have

$$\begin{aligned}
\chi_{xy}(t) &= \langle x_c(t)y_q(0)\rangle \simeq U_{11}U_{21}\langle\phi_c(t)\phi_q(0)\rangle \,, \\
\chi_{yx}(t) &= \langle y_c(t)x_q(0)\rangle \simeq U_{12}U_{11}\langle\phi_c(t)\phi_q(0)\rangle \,.
\end{aligned} \qquad (45)$$

Again using the fact that $U_{12} = -U_{21}$, one can see that $\chi_{xy}(t) \simeq -\chi_{yx}(t)$.

Finally, we remark that the field $y$ becomes gapped at the weakly dissipative point as one can see from Eq. (41) (see also Fig. 4(b)), which leads to the noncritical $\langle\hat{S}_y^2\rangle$ fluctuations. One thus recovers the equilibrium behavior although one should take the limit $\Gamma \to 0$ with care due to the order of limits discussed in Section 4.3.

## 5.2 Onsager reciprocity relations

In this section, we derive the modified form of the Onsager reciprocity relations. As a starting point, consider the saddle-point solution of Eq. (37): $\mathbf{D}^R(i\partial_t) \cdot \mathbf{x}(t) = 0$ where we have replaced $\omega \to i\partial_t$ and defined $\mathbf{x} = (x, y)$; we have dropped the subscript $c$ for convenience. By rearranging the time derivatives, we find the equation

$$\frac{\mathrm{d}}{\mathrm{d}t}\mathbf{x}(t) = -\mathbf{M} \cdot \mathbf{x}(t)\,, \qquad \text{with} \quad \mathbf{M} = \begin{pmatrix} \Gamma/2 & 2\Delta \\ 4J - 2\Delta & \Gamma/2 \end{pmatrix}. \qquad (46)$$

This equation describes the average dynamics of $\mathbf{x}(t)$ (i.e., $\langle\hat{S}_{x,y}\rangle$) near the steady state and governs its decay to zero.

Adopting a slightly more general notation, the dynamics near the steady state can be written as

$$\frac{\mathrm{d}}{\mathrm{d}t}\langle x_i\rangle_t = -\sum M_{ik}\langle x_k\rangle_t, \qquad (47)$$

where $\{x_i\}$ denote a set of macroscopic variables, and $\langle\cdot\rangle_t$ represents the statistical (and, the quantum) average at time $t$; we later specialize to the variable $\mathbf{x}$ by setting $x_1 \equiv x$ and $x_2 \equiv y$. Now defining $L_{ij} = \sum_k M_{ik}\langle x_k x_j\rangle$, Onsager reciprocity relations in equilibrium take the form

$$L_{ij} = \epsilon_i\epsilon_j L_{ji}\,, \qquad (48)$$

where $\epsilon_i$ denotes the parity of the corresponding field under time-reversal transofrmation. These relations are a direct consequence of the equilibrium FDR—in the form of Onsager's regression hypothesis—together with the TRS. The Onsager reciprocity relations are of great importance for their fundamental significance as well as practical applications. We shall refer the interested reader to Ref. [33] for the proof of the reciprocity relations in a classical setting.

In the non-equilibrium context of our model with both FDR and TRS broken, the Onsager reciprocity relations do not generally hold; however, given the modified form of the FDR* and TRS* in Eqs. (5) and (6), one may expect a modified form of the Onsager relations perhaps with a different parity than the one expected in equilibrium. Here, we show that this is indeed the case. To this end, we first note that the Onsager's regression hypothesis is modified in a straightforward fashion as

$$\langle x_i \rangle_t = \epsilon_j \frac{\lambda}{k_B T} \langle x_i(t) x_j(0) \rangle, \tag{49}$$

assuming that a "magnetic" field $\lambda$ has been applied along the $i$ direction before it is turned off at time $t = 0$. The only difference from the standard Onsager regression hypothesis is the prefactor $\epsilon_j$ appearing out in front, a factor that simply carries over from Eq. (5). Combining with Eq. (47), we have

$$\frac{\mathrm{d}}{\mathrm{d}t} \langle x_i(t) x_j(0) \rangle = -\sum M_{ik} \langle x_k(t) x_j(0) \rangle. \tag{50}$$

Notice that the factors of $\epsilon_j$ cancel out on both sides. Finally, using the TRS* of the correlation function, $C_{ij}(t) = C_{ji}(t)$ regardless of the corresponding parities, and setting $t = 0$, we find[3]

$$L_{ij} \simeq L_{ji}. \tag{51}$$

Notice the absence of the TRS parity factors $\epsilon_i \epsilon_j$; cf. the equilibrium Onsager reciprocity relation in Eq. (48).

To verify that this relation holds in our non-equilibrium setting, it is important to distinguish the contribution of the soft mode, responsible for the critical behavior, from the gapped mode. Therefore, we shall consider the dynamics at a coarse-grained level where the gapped mode is "integrated out". To this end, let's write

$$\mathbf{M} = m \boldsymbol{\phi}^R \boldsymbol{\phi}^L + M \boldsymbol{\zeta}^R \boldsymbol{\zeta}^L, \tag{52}$$

where we have used a dyadic notation. Here, $\boldsymbol{\phi}^{R/L}$ and $\boldsymbol{\zeta}^{R/L}$ define the right/left eigenvectors of the matrix $\mathbf{M}$. These vectors are biorthogonal, that is, $\boldsymbol{\phi}^L \cdot \boldsymbol{\phi}^R = \boldsymbol{\zeta}^L \cdot \boldsymbol{\zeta}^R = 1$ while $\boldsymbol{\phi}^L \cdot \boldsymbol{\zeta}^R = \boldsymbol{\zeta}^L \cdot \boldsymbol{\phi}^R = 0$. Furthermore, $m$ and $M$ represent the two eigenvalues of the matrix $\mathbf{M}$: the eigenvalue $m$ vanishes at the critical point defining the soft mode, while $M$ remains finite (at the order of $J$) and defines the gapped mode. The notation for the soft and gapped modes mirror our conventions for the effective field theory. In fact, the above diagonalization is a similar decomposition to that of the previous section but in a different basis (notice that $\mathbf{M}$ is "rotated" with respect to $\mathbf{D}^R$). While we do not need the explicit form of the eigenvalues and the (right and left) eigenvectors, here we provide

---

[3]Since the modified FDR doesn't hold at short times, setting $t = 0$ might seem problematic. However, the error incurred in the process only amounts to a noncritical correction.

them for completeness:

$$\phi^R = \left(-\sqrt{\frac{\Delta}{2J - \Delta}}\,,1\right), \quad \phi^L = \frac{1}{2}\left(-\sqrt{\frac{2J - \Delta}{\Delta}}\,,1\right), \quad m = \Gamma - 4\sqrt{(2J - \Delta)\Delta}\,,$$

$$\zeta^R = \left(\sqrt{\frac{\Delta}{2J - \Delta}}\,,1\right), \quad \zeta^L = \frac{1}{2}\left(\sqrt{\frac{2J - \Delta}{\Delta}}\,,1\right), \quad M = \Gamma + 4\sqrt{(2J - \Delta)\Delta}\,.$$

$$(53)$$

Now, the coarse-grained dynamics at sufficiently long times is governed solely by the soft mode, while the gapped field quickly decays to zero ($\zeta^L \cdot \mathbf{x} = 0$). Therefore, the slow dynamics is given by

$$\frac{\mathrm{d}}{\mathrm{d}t}\mathbf{x} = -\overline{\mathbf{M}} \cdot \mathbf{x}\,, \tag{54}$$

where we have defined $\overline{\mathbf{M}} = m\boldsymbol{\phi}^R\boldsymbol{\phi}^L$ keeping only the critical component. We are finally in a position to study the relation between $L_{xy}$ and $L_{yx}$ explicitly defined by

$$L_{xy} = \overline{M}_{xx}\langle xy\rangle + \overline{M}_{xy}\langle yy\rangle\,,$$
$$L_{yx} = \overline{M}_{yx}\langle xx\rangle + \overline{M}_{yy}\langle yx\rangle\,. \tag{55}$$

Now notice that the fluctuations $\langle x_i x_j\rangle \sim \phi_i^R \phi_j^R \langle\phi^2\rangle$ where $\langle\phi^2\rangle$ represents the critical fluctuations (to be identified with $\langle\phi_c^2\rangle$ in the previous section); this simply means that the dominant contribution to fluctuations is given by the overlap of dynamical variables with the critical field. Additionally, using the biorthogonality $\zeta^L \cdot \phi^R = 0$, we have $\left(\zeta_1^L, \zeta_2^L\right) \propto \left(-\phi_2^R, \phi_1^R\right)$. We can then write

$$L_{xy} - L_{yx} \propto \zeta^L \cdot \overline{\mathbf{M}} \cdot \phi^R = 0\,, \tag{56}$$

where the last equality follows from $\zeta^L \cdot \overline{\mathbf{M}} \propto \zeta^L \cdot \phi^R = 0$.[4] We thus arrive at the relation $L_{yx} \simeq L_{xy}$ in harmony with our modified version of the Onsager reciprocity relation. This should be contrasted with the reciprocity relation in equilibrium: $L_{xy} = -L_{yx}$ with $x$ ($y$) even (odd) under time-reversal transformation.

## 6 Driven-Dissipative Coupled Bosons

In this section, we go beyond the infinite-range model discussed so far and consider a quadratic model of driven-dissipative bosons. The model being quadratic can be solved exactly using any number of techniques. For a coherent presentation, we will adopt a simple (Keldysh) field-theoretical analysis. Our main point is however that the conclusions of this work apply to a wider range of models. To be specific, consider a bosonic model on a cubic lattice in $d$ dimensions with the Hamiltonian

$$\hat{H} = -\frac{J}{2d}\sum_{\langle\mathbf{ij}\rangle}(\hat{a}_{\mathbf{i}} + \hat{a}_{\mathbf{i}}^\dagger)(\hat{a}_{\mathbf{j}} + \hat{a}_{\mathbf{j}}^\dagger) + 2\Delta\sum_{\mathbf{i}}\hat{a}_{\mathbf{i}}^\dagger\hat{a}_{\mathbf{i}}\,, \tag{57}$$

and subject to the dissipation

$$\hat{L}_{\mathbf{i}} = \sqrt{\Gamma}\,\hat{a}_{\mathbf{i}}\,. \tag{58}$$

The coefficients in the Hamiltonian are chosen for later convenience. Notice that the Hamiltonian is time-reversal symmetric. This follows from either writing the operator $\hat{a}$

---

[4]While one might be tempted to conclude that $\overline{\mathbf{M}} \propto m \to 0$ at the critical point, the product $m\langle\phi^2\rangle$ remains finite due to the diverging fluctuations and thus $L_{xy}$ assumes a nonzero value at the critical point.

in terms of two quadratures that are even and odd under time-reversal (see below), or directly by noting that $\hat{T}\hat{a}\hat{T}^{-1} = \hat{a}$ and similarly for $\hat{a}^\dagger$ (site index suppressed) although $\hat{T}$ is antiunitary ($\hat{T}i\hat{T}^{-1} = -i$) [55]. The above bosonic Hamiltonian is therefore real and time-reversal symmetric.

The Keldysh action for this model can be constructed in a straightforward fashion using a coherent-state representation mapping operators to c-valued fields as $\hat{a}_{\mathbf{i}} \to a_{\mathbf{i}}(t)$ and $\hat{a}_{\mathbf{i}}^\dagger \to a_{\mathbf{i}}^*(t)$. A path-integral formalism can be straightforwardly constructed in terms of these bosonic fields on a closed contour with the Keldysh action given by [20]

$$S_K = S_H + S_D, \tag{59}$$

where $S_{H,D}$ represent the coherent and dissipative terms, respectively. The coherent term in the action is given by

$$S_H = \sum_{\sigma=+,-} \sigma \int_t \Big[ \sum_{\mathbf{i}} a_{\mathbf{i}\sigma}^* i\partial_t a_{\mathbf{i}\sigma} - H[a_{\mathbf{i}\sigma}, a_{\mathbf{i}\sigma}^*] \Big], \tag{60}$$

with $\sigma = \pm$ representing the forward and backward branches of the contour. The last term represents the (normal-ordered) Hamiltonian in the coherent-state representation. The relative sign of the forward and backward branches has its origin in the commutator $[\hat{H}, \hat{\rho}]$. The dissipative term in the action takes the form

$$S_D = -i\Gamma \sum_{\mathbf{i}} \int_t \Big[ a_{\mathbf{i}+} a_{\mathbf{i}-}^* - \frac{1}{2}\big( a_{\mathbf{i}+}^* a_{\mathbf{i}+} + a_{\mathbf{i}-}^* a_{\mathbf{i}-} \big) \Big]. \tag{61}$$

Upon a Keldysh rotation $a_{cl/q} \equiv (a_+ \pm a_-)/\sqrt{2}$ (site index $\mathbf{i}$ being implicit), the Keldysh action is then written in terms of classical and quantum fields. Here, it is more convenient to cast the bosonic field in terms of its real and imaginary parts (the two quadratures) as $a_{\mathbf{i}}(t) = (\Phi_{\mathbf{i}}(t) - i\Pi_{\mathbf{i}}(t))/2$ where the factor of $1/2$ is chosen for later convenience. The corresponding operators can be viewed as a scalar field and the conjugate momentum. These Hermitian operators obey the same symmetry relations as $x$ and $y$ in the DDIM, where $\Phi$ is even under TRS while $\Pi$ is odd. The anti-unitary nature of the time-reversal transformation makes the bosonic fields real and invariant under TRS. The Lagrangian $L_K$ defined via the Keldysh action $S_K = \int dt L_K$ then takes the form [20]

$$L_K = \sum_{\mathbf{i}} \frac{1}{2}\Phi_{\mathbf{i}q}\partial_t \Pi_{\mathbf{i}c} - \frac{1}{2}\Pi_{\mathbf{i}q}\partial_t \Phi_{\mathbf{i}c} - \Delta(\Phi_{\mathbf{i}c}\Phi_{\mathbf{i}q} + \Pi_{\mathbf{i}c}\Pi_{\mathbf{i}q}) + \frac{\Gamma}{4}(\Phi_{\mathbf{i}q}\Pi_{\mathbf{i}c} - \Phi_{\mathbf{i}c}\Pi_{\mathbf{i}q} + i\Phi_{\mathbf{i}q}^2 + i\Pi_{\mathbf{i}q}^2)$$
$$+ \sum_{\langle \mathbf{i}\mathbf{j} \rangle} \frac{J}{2d}(\Phi_{\mathbf{i}c}\Phi_{\mathbf{j}q} + \Phi_{\mathbf{i}q}\Phi_{\mathbf{j}c}), \tag{62}$$

in terms of classical and quantum fields $\Phi_{\mathbf{i}c/q}$ and $\Pi_{\mathbf{i}c/q}$. In momentum space, the Keldysh action takes almost an identical form to Eq. (37) with the substitution $v \to (\Phi_c, \Pi_c, \Phi_q, \Pi_q)$ where the frequency and momentum $(\omega, \mathbf{k})$ are implicit and $J \to J_{\mathbf{k}} = \frac{J}{d}(\cos k_1 + \cdots + \cos k_d)$. This implies that this model too exhibits a phase transition at the same set of parameters. While a nonlinear term is needed to regulate things on the ordered side, we shall only consider the critical behavior.

## 6.1 Green's functions

Since Eq. (62) is identical to Eq. (37) upon the above substitutions, we can immediately write the correlation and response functions of $\Phi$ and $\Pi$. They are simply given by Eq. (23)

once with $J$ is substituted by $J_{\mathbf{k}}$. Using the definitions of the bosonic variables in terms of the real fields, we can easily determine the form of the bosonic Green's functions:

$$\mathbf{G}^K = \begin{pmatrix} G_{aa^\dagger}^K & G_{aa}^K \\ G_{a^\dagger a^\dagger}^K & G_{a^\dagger a}^K \end{pmatrix} , \quad \mathbf{G}^R = \begin{pmatrix} G_{aa^\dagger}^R & G_{aa}^R \\ G_{a^\dagger a^\dagger}^R & G_{a^\dagger a}^R \end{pmatrix} , \tag{63}$$

where

$$G_{aa^\dagger}^K(\omega, \mathbf{k}) = \left[ G_{a^\dagger a}^K(-\omega, \mathbf{k}) \right]$$
$$= \frac{-i\Gamma \left( 3\Gamma^2 + 4(32J_{\mathbf{k}}^2 + 12\Delta^2 + 8\Delta\omega + 3\omega^2 - 8J_{\mathbf{k}}(4\Delta + \omega)) \right)}{8(\omega - \omega_1)(\omega - \omega_2)(\omega - \omega_1^*)(\omega - \omega_2^*)} , \tag{64a}$$

$$G_{aa}^K(\omega, \mathbf{k}) = - \left[ G_{a^\dagger a^\dagger}^K(\omega, \mathbf{k}) \right]^*$$
$$= \frac{i\Gamma \left( 128J_{\mathbf{k}}^2 + \Gamma^2 - 16iJ_{\mathbf{k}}(\Gamma - 8i\Delta) + 4(4\Delta^2 + \omega^2) \right)}{8(\omega - \omega_1)(\omega - \omega_2)(\omega - \omega_1^*)(\omega - \omega_2^*)} , \tag{64b}$$

$$G_{aa^\dagger}^R(\omega, \mathbf{k}) = \left[ G_{a^\dagger a}^R(-\omega, \mathbf{k}) \right]^* = \frac{-4J_{\mathbf{k}} + 4\Delta + 2\omega + i\Gamma}{2(\omega - \omega_1)(\omega - \omega_2)} , \tag{64c}$$

$$G_{aa}^R(\omega, \mathbf{k}) = \left[ G_{a^\dagger a^\dagger}^R(\omega, \mathbf{k}) \right]^* = \frac{-2(J_{\mathbf{k}} - \Delta)}{(\omega - \omega_1)(\omega - \omega_2)} , \tag{64d}$$

and $\mathbf{G}^R(\omega, \mathbf{k}) = [\mathbf{G}^A(\omega, \mathbf{k})]^\dagger$, $\mathbf{G}^K(\omega, \mathbf{k}) = -[\mathbf{G}^K(\omega, \mathbf{k})]^\dagger$. In a slight abuse of notation, we have defined the modes $\omega_{1/2} = -i(\Gamma \mp \Gamma_c(J_{\mathbf{k}}))/2$ (introduced earlier in Section 3.2) and defined the function $\Gamma_c(J) \equiv 4\sqrt{\Delta(2J - \Delta)}$.

For comparison with the FDR in the time-domain, we quote the long-wavelength ($\mathbf{k} \to 0$) limit of the correlation and response functions at criticality:

$$G_{aa^\dagger}^K(t, \mathbf{k}) = G_{a^\dagger a}^K(-t, \mathbf{k}) \sim \frac{-i4dJ}{\Delta\mathbf{k}^2} e^{-A\mathbf{k}^2|t|} , \tag{65a}$$

$$G_{aa}^K(t, \mathbf{k}) = - \left[ G_{a^\dagger a^\dagger}^K(t, \mathbf{k}) \right]^* \sim \frac{4d}{\mathbf{k}^2} \left[ \frac{-i(J + \Delta)}{\Delta} + \frac{4(2J - \Delta)}{\Gamma_c} \right] e^{-A\mathbf{k}^2|t|} , \tag{65b}$$

$$G_{aa^\dagger}^R(t, \mathbf{k}) = \left[ G_{a^\dagger a}^R(t, \mathbf{k}) \right]^* \sim \Theta(t) \left( \frac{8(J - \Delta)}{\Gamma_c} - 2i \right) e^{-A\mathbf{k}^2 t} , \tag{65c}$$

$$G_{aa}^R(t, \mathbf{k}) = \left[ G_{a^\dagger a^\dagger}^R(t, \mathbf{k}) \right]^* \sim \Theta(t) \frac{-8J}{\Gamma_c} e^{-A\mathbf{k}^2 t} , \tag{65d}$$

where we have defined $A = -J\Gamma_c/4d(2J - \Delta)$ and $\Gamma_c = \Gamma_c(J)$. The expressions above are obtained by first setting $\Gamma = \Gamma_c$ and then taking the limit $\mathbf{k} \to 0$ while keeping $\mathbf{k}^2 t = \text{const}$. These expressions are valid all along the phase boundary except at the weakly-dissipative critical point since we have assumed $\mathbf{k}^2 \ll 2J - \Delta$ in our derivation.

## 6.2  FDR* for non-Hermitian operators

The Green's functions of $\Phi$ and $\Pi$ of the short-range model considered here are identical to those of the DDIM once we substitute $J \to J_{\mathbf{k}}$. Therefore, the low-frequency effective temperatures of this model in the long-wavelength limit $\mathbf{k} \to 0$ are *identical* to those of the DDIM in Eq. (29). In other words, at criticality and at long wavelengths this short-ranged model obeys the FDR*. The latter can be extended to the bosonic operators $\hat{a}_{\mathbf{k}}$ and $\hat{a}_{\mathbf{k}}^\dagger$ too. Taking the linear combination of the FDR* for the two quadratures, we find

$$\chi_{a_{\mathbf{k}}^\dagger a_{\mathbf{k}}} \simeq \frac{1}{2T_{\text{eff}}} \Theta(t) \partial_t C_{a_{-\mathbf{k}}^\dagger a_{\mathbf{k}}^\dagger} , \qquad \chi_{a_{\mathbf{k}} a_{-\mathbf{k}}} \simeq \frac{1}{2T_{\text{eff}}} \Theta(t) \partial_t C_{a_{\mathbf{k}} a_{\mathbf{k}}^\dagger} . \tag{66}$$

These relations can be explicitly verified by plugging in Eq. (65) with the effective temperature identified as $T_{\text{eff}} = J$. Interestingly, the set of operators on the two sides of these FDR-like equations are different, namely the first operator (appearing at the earlier time) transforms into its adjoint between the two sides of these equations.

The above equation suggests a more general form of the FDR*, also applicable to non-Hermitian operators, as

$$\chi_{O_i O_j^T}(t) \simeq \frac{1}{2T_{\text{eff}}} \Theta(t) \partial_t C_{O_i O_j} \,, \tag{67}$$

where $\hat{O}_i$s are not necessarily Hermitian. The transpose $T$ arises due to the combined action of taking the adjoint as well as conjugation due to the time-reversal transformation. This equation reduces to the FDR* for Hermitian operator in Eq. (4), while reproducing Eq. (66) for non-Hermitian (but real) bosonic operators.

## 6.3 Weakly-dissipative limit

Finally, we investigate the bosonic Green's functions at the weakly-dissipative point; this parallels our discussion of the weakly-dissipative DDIM in Section 4.3. Again we must be careful in taking the order of limits. We shall first Fourier transform Eq. (64) to the time domain, send $\Delta \to 2J$, and then take the long-wavelength limit $\mathbf{k} \to 0$ in which case we have $J_{\mathbf{k}} \sim J(1 - \mathbf{k}^2/2d)$ and $\Gamma_c(J_{\mathbf{k}}) \sim i4\sqrt{2}J|\mathbf{k}|/d$. Finally, we take the weakly-dissipative limit $\Gamma \to 0$ and report only the critical contribution at long wavelengths:

$$G_{\alpha\beta}^K(t, \mathbf{k}) \sim -i\frac{2d^2}{\mathbf{k}^2} \cos\left(\frac{2\sqrt{2}J}{d}|\mathbf{k}|t\right) \,, \tag{68a}$$

$$G_{\alpha\beta}^R(t, \mathbf{k}) \sim -\Theta(t)\frac{2\sqrt{2}d}{|\mathbf{k}|} \sin\left(\frac{2\sqrt{2}J}{d}|\mathbf{k}|t\right) \,, \tag{68b}$$

for $\alpha, \beta \in \{a, a^\dagger\}$. Note that the dynamical exponent ($z$) is now different as the scaling variable is $|\mathbf{k}|t$ compared to $\mathbf{k}^2 t$ in Eq. (65), i.e., we find ballistic ($z = 1$) rather than diffusive dynamics ($z = 2$). Fluctuations diverge in the same fashion, $G_{\alpha\beta}^K \sim 1/\mathbf{k}^2$, regardless of the dissipation, while the dynamical behavior undergoes a crossover; for a similar behavior of the DDIM, see Ref. [38]. As we kept $\mathbf{k}$ finite while taking $\Gamma \to 0$, the system remains gapped. Therefore, the density matrix commutes with the Hamiltonian, in parallel with our discussion in Section 4.3. The TRS is then restored and the correlation and response functions satisfy the equilibrium FDR as one can directly see from Eq. (68). If we instead take $\mathbf{k} \to 0$ before sending $\Gamma \to 0$, we find that the cross-correlation $G_{\Phi\Pi}^K(t = 0, \mathbf{k} = 0) \sim 1/\Gamma$ diverges even at the weakly-dissipative critical point, while this quantity remains zero in equilibrium as it is odd under the time-reversal transformation.

# 7 Conclusion and Outlook

In this work, we have considered Ising-like driven-dissipative systems where the Hamiltonian itself is time-reversal symmetric although dissipation breaks this symmetry. We have shown that, despite an emergent effective temperature, the FDR and TRS are macroscopically violated when one considers multiple operators that overlap with the order parameter and are even or odd under time-reversal transformation. Nevertheless, we have argued that a modified form of the fluctuation-dissipation relation (dubbed FDR*) governs the critical behavior. Similarly, a modified form of time-reversal symmetry (dubbed TRS*) arises

where correlation and response functions find definite, but possibly opposite, parities under time-reversal transformation; in sharp contrast with TRS in equilibrium, one cannot assign a well-defined parity to a given operator while correlation and response functions exhibit definite parities. Additionally, we have derived a modified form of the Onsager reciprocity relation in harmony with the TRS* while violating the TRS. These conclusions are based on the underlying symmetries (time-reversal symmetry of the Hamiltonian and the Ising symmetry of the full Liouvillian) and the existence of a single soft mode at the phase transition. They follow from a generic field-theoretical analysis that leads to a non-Hermitian kernel for the dynamics. We have presented our results in the context of two relatively simple Ising-like driven-dissipative systems. Finally, we have shown that even in the limit of vanishing dissipation, TRS is not necessarily restored.

We distinguish our results from recent interesting works where quantum detailed balance, microreversibility and time-reversal symmetry [56–58] or extensions thereof [59] are an exact property of a special class of open quantum systems. On the other hand, the modified time-reversal symmetry of two-time correlators introduced here arises near criticality, but is expected to hold for a large class of driven-dissipative systems near their phase transitions. More generally, the fundamental nature of time-reversal and its far-reaching consequences has brought it to the center stage of research on open quantum systems.

An natural extension of the models considered here is the full Dicke model with both bosonic and spin operators in a single- or multi-mode cavity [30, 41, 60]. The two components of the the spin operators as well as the two quadratures of the cavity mode(s) constitute a larger space of operators that overlap with the order parameter and are even/odd under time-reversal transformation, but similar results should be expected. Another interesting direction is to go beyond mean-field or quadratic models and consider nonlinear interactions and their effect on the modified fluctuation-dissipation relations and time-reversal symmetry. An important future direction is to investigate if similar FDR* and TRS* emerge for phase transitions governed by different symmetries. It is possible that a generalization of the results reported in this work would depend on the underlying symmetries, as well as the weak or strong nature of such symmetries [61]. Similarly, one may consider models where the time-reversal transformation takes a more complicated form than complex conjugation. More generally, it is desired to identify emergent forms of time-reversal symmetry governing the the macroscopic behavior of driven-dissipative systems although this symmetry is generically broken microscopically.

**Funding information**   This material is based upon work supported by the NSF under Grant No. DMR-1912799. M.M. also acknowledges support from the Air Force Office of Scientific Research (AFOSR) under award number FA9550-20-1-0073 as well as the start-up funding from Michigan State University.

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
