# Peer review of "Time-reversal symmetry breaking and resurrection in driven-dissipative Ising models"

_SciPost Physics_

## Round 1 · Referee Report · Dominic Rose (Referee 1) · 2021-7-27

Strengths

  1. Provides an excellent contribution to current research in the emergence and impact of time reversal-like symmetries and fluctuation dissipation relations in open quantum systems.
  2. Calculations are clearly presented.
  3. The work is well motivated and summarised.

Weaknesses

While the results are impressive, at present, several are conveyed in such a way that they sound more general than they appear to be.

Report

In this contribution the authors consider the question of how time-reversal symmetry (TRS) and fluctuation dissipation relations (FDR) can emerge from open quantum system dynamics. They outline how a central model for the study of open quantum systems, the fully-connected driven-dissipative Ising model, does not exhibit either TRS or FDR, despite work suggesting such properties are emergent near phase-transitions in similar systems. Instead, they show that for a subset of observables, a modified set of TRS and FDR-like relationships between correlation and response functions can be found. To further demonstrate the relevance of this behaviour, a second model is considered which features finite-range quadratic interactions between bosons on a cubic lattice, exhibiting similar equations.

I believe this is a groundbreaking set of result which highlights the subtleties of whether TRSs or FDRs do infact emerge from non-equilibrium systems, demonstrating that they may only occur for a limited set of observables. It further demonstrates that by broadening what form we allow a TRS or FDR to take, these relationships may be extended to a broader, if still limited, set of observables. The work presents a range of future directions for study, such as the role of nonlinear interactions, and a more general understanding of the conditions under which such modified relations occur.

I therefore believe it meets the expectations required by this journal. The presentation is of a high quality and generally clear, meeting the general acceptance criteria of the journal.

I note that while very well written, the current presentation could cause some confusion in the understanding of the results and the generality of their applicability. Addressing these points could help other researchers more easily understand how to build on this work in the future. Specifically:

  1. The choice is made to conduct a variety of calculation steps under the assumption of the modified TRS, rather than for the specific models considered. While this demonstrates how a certain results may extend to other models with similar properties, it can cause some confusion. In particular, this makes it appear as though the effective temperatures in Eq. (29) are valid for a broad range of parameters, despite being derived using the modified TRS which is, in the next section, shown to hold only near the phase transition.

  2. In standard equilibrium situation we would expect the TRS and FDR to hold for all observables. In section 2 it is not made particularly clear that the modified relations only hold for a subset of observables, only that the standard relations can be shown to fail by using specific observables. For example, below Eq. 4 it merely says the observables are hermitian. Similarly, there is no mention around Eq. (6) that it only holds for a subset of observables. I note that this is stated more clearly in the introduction already.

  3. The abstract / introduction / section 2 appear to imply that that the results found apply generally to driven-dissipative Ising-type systems. Despite this, all calculations appear to be for two specific models. While this may be suggestive, it does not seem to imply this is a generic behaviour of all similar models. More discussion of the reasons for why it may be believed to hold more generally could be benefitial.

  4. The general statement of the results appears to be motivated by the presence of the $\mathbb{Z}_2$ symmetry. Despite this, it is not particularly clear why this symmetry is needed to arrive at the results. Indeed, no such symmetry is mentioned in section 6. This is in contrast to the importance of the TRS of the Hamiltonian, which is explicitly and clearly discuss below Eq. 43.

While these points detract from the overall clarity of the results, I do not believe they detract from the high quality / relevance of the research.

Requested changes

  1. Please specify explicitly that $\Theta(t)$ is the Heaviside step function.
  2. Please clarify the applicability of the modified TRS and FDR relations in section 2.
  3. Please define $\mathcal{F}_\omega$ between Eqs. 22 and 23.
  4. Typo: Eq. 5 is referenced in the second paragraph of section 4.1, when it appears to mean Eq. 6.
  5. Please clarify that while Eq. 29 has been calculated throughout the normal phase, they rely on the modified TRS that is valid only near the phase transition.
  6. Typo: Second sentence of section 5.1, "different sets of operators".
  7. It is not clear why the gapped field is not relevant to the effective temperature in section 5.1. Is it due to the long-wavelength limit? If so, the order of statements seems off. Likewise for Eqs. 44 and 45. Please clarify why this can be discarded.
  8. Typo: The last paragraph of the conclusion should start with "A".
  9. Please clarify the relevance of the $\mathbb{Z}_2$ symmetry to the calculations.
  10. Please clarify that the results are suggestive of what may hold for generic driven-dissipative Ising-type systems, or otherwise justify this generic statement.

---

## Round 1 · Referee Report · Anonymous (Referee 2) · 2021-7-28

Strengths

1) Provides a generalisation of the ideas of time reversal symmetry to non-equilibrium steady states 2) Clearly explained calculations with well chosen examples

Weaknesses

1) In some places the physical intuition for the results could be expanded

Report

The paper by Paz and Magrhebi presents a generalisation of the ideas of time reversal symmetry to systems in a non-equilibrium steady state. They show how this leads to a slightly modified form of the fluctuation dissipation relation which includes an extra minus sign depending on the symmetry of the operators considered. These results are mostly clearly presented and give a lot of insights into the behaviour of systems driven out-of-equilibrium. They will be useful to a wide community of theorists working on related topics and so I recommend publication in SciPost Physics.

Requested changes

There are a few points that the authors should address to aid in clarifying their arguments in a few points in the manuscript:

1) It would possibly be useful to also introduce the Fourier space versions of eqns 4-6 as these could be more familiar to some readers. 2) Should eqn 10 read |\rho(0)>>? 3) It could be useful to give a few lines more explanation of the physical intuition behind eqn 15. 4) In Fig 2 is the disagreement at short times due to finite system size? It may be useful to show a couple of values of N here to illustrate this. 5) In the discussion around eqn 35 the authors find that the two ways of taking the limit \Gamma -> 0 give different results. How does this behaviour show up in finite sized numerics? Is there a smooth change between the different parities? 6) It would be useful for the authors to add a few more comments to the conclusions about how general they think the results obtained here are. Does any Markovian model with a phase transition show this kind of FDR?

  • validity: high
  • significance: high
  • originality: high
  • clarity: good
  • formatting: good
  • grammar: excellent

Author:  Daniel Paz  on 2021-09-21  [id 1770]

(in reply to Report 2 on 2021-07-28)
Category:
answer to question

In this reply, we supplement our response to the Referee's 4th point regarding Fig. 2. For our complete response to all of the comments, please see the resubmission change list.

The disagreement at short times is actually due to the applicability of the FDR. As stated in the derivation, Eq. (5) only is valid at long times/low frequencies. This leads to the short-time disagreement. In fact, it is rather surprising how quickly the convergence occurs and the slow, critical dynamics sets in. We have included a few words emphasizing the origin of the disagreement in the last paragraph of Sec. 4.2 .

In addition, we have attached to this comment a figure showing the FDR relation for different system sizes. We find that the time when the two curves begin to agree with different system sizes does not move left as system size is increased, indicating that the disagreement is not primarily due to finite-size corrections. The disagreement exists even in the thermodynamic limit, as the exact expressions for the correlation and response function contain non-critical contributions at short times. Only at low-frequencies and by neglecting the non-critical contributions, do we find that they obey the emergent FDR.

Attachment:

FDR_vstretched_shorttime.pdf

---

## Round 1 · Referee Report · Anonymous (Referee 3) · 2021-7-29

Strengths

1- new important results 2- quite well written 3- important subject

Weaknesses

1- slightly technical at times 2- some aspects could be better justified and some presentation can be improved

Report

In this paper, the authors propose a modification of the time-reversal symmetry and fluctuation-dissipation relations that are known to hole at equilibrium. The modification is proposed to describe open quantum systems in non-equilibrium steady states, and is supposed to hold at criticality. The relation is verified in the driven-dissipative Ising model (where atoms are emitted at a given rate, that is spins ``fall down”), and in a quadratic boson models, and is argued for by a mean-field analysis based on the Keldysh formulation and a Hubbard-Stratonovich transformation.

The paper is very well written, the result well argued, the derivations are somewhat technical but perfectly fine and seem to be correct. The model is important and the question of characterising non-equilibrium steady states is a question receiving a large amount of attention in modern research. I believe this paper present a very interesting development with interesting ideas. I am happy to accept the paper for publication; I propose minor improvements in the discussion below.

1) There is a commutator in definition of response, valid in quantum systems, but Eq 2 is said to be valid in classical systems as well; please clarify the definition of response function for classical systems.

2) Eq 10: should $\rho_{ss}$ be replaced by $\rho_0$, and the limit $t\to\infty$ be taken? $\rho_{ss}$ should be invariant under the evolution operator.

3) Eq 24 is valid in limit $\omega\to0$ only, similarly Eq 2 is valid as $\hbar\to0$; is this what is meant by saying this holds only at large times? My understanding is that the authors look for large times because such correlations / response functions are controlled by the soft mode, hence this characterises criticality. Is this correct? Otherwise, why look only at the long-time dynamics? How would other modes affect FDR and TRS? Would they contribute to higher powers of $\omega$? The equilibrium FDR includes a non-linear function of $\omega$, could one see differences also for $\omega$ further from 0 and attribute this to the soft modes?

4) I don’t understand the use of the symbol $\simeq$ on page 9: are these equalities? If not, in what sense are the left-hand and right-hand sides related, is it in a limit sense? What limit - is it the long-time limit?

5) For clarity, please do say where the phase transitions is ($\Gamma = \Gamma_c$ expressed in terms of $\Delta$ and $J$?) early enough in the discussion.

6) Finally, why “resurrection” in the title? It seems nowhere else in the paper the concept of resurrection is mentioned.

Requested changes

1- clarify definition of classical response function 2- clarify eq 10 3- clarify the need to take lont times / small frequencies 4- clarify \simeq on page 9 5- write phase transition point earlier in paper 6- discuss "resurrection" mentioned in title, or take away from title.

---

## Editorial Decision

resubmitted